# Deregulation of the Interleukin-7 Signaling Pathway in Lymphoid Malignancies

**DOI:** 10.3390/ph14050443

**Published:** 2021-05-08

**Authors:** Inge Lodewijckx, Jan Cools

**Affiliations:** 1Center for Human Genetics, KU Leuven, Herestraat 49, 3000 Leuven, Belgium; inge.lodewijckx@kuleuven.be; 2Center for Cancer Biology, VIB, Herestraat 49, 3000 Leuven, Belgium; 3Leuven Cancer Institute (LKI), KU Leuven/UZ Leuven, Herestraat 49, 3000 Leuven, Belgium

**Keywords:** interleukin-7, cytokine receptor, JAK kinases, signaling, lymphocyte development, lymphoid malignancy, acute lymphoblastic leukemia, kinase inhibitor, targeted treatment

## Abstract

The cytokine interleukin-7 (IL-7) and its receptor are critical for lymphoid cell development. The loss of IL-7 signaling causes severe combined immunodeficiency, whereas gain-of-function alterations in the pathway contribute to malignant transformation of lymphocytes. Binding of IL-7 to the IL-7 receptor results in the activation of the JAK-STAT, PI3K-AKT and Ras-MAPK pathways, each contributing to survival, cell cycle progression, proliferation and differentiation. Here, we discuss the role of deregulated IL-7 signaling in lymphoid malignancies of B- and T-cell origin. Especially in T-cell leukemia, more specifically in T-cell acute lymphoblastic leukemia and T-cell prolymphocytic leukemia, a high frequency of mutations in components of the IL-7 signaling pathway are found, including alterations in *IL7R*, *IL2RG*, *JAK1*, *JAK3*, *STAT5B*, *PTPN2*, *PTPRC* and *DNM2 genes*.

## 1. The Role of IL-7 Signaling in Lymphocyte Development

Interleukin-7 (IL-7) is crucial for lymphocyte development, as well as for the survival, proliferation, differentiation and activity of mature T- and B-cells [1]. The IL-7 receptor (IL-7R) is a heterodimer consisting of the specific IL-7Ralpha chain (IL-7Rα, CD127; encoded by the *IL7R* gene) and the common gamma chain (γc, CD132; encoded by the *IL2RG* gene), which is shared by receptors for IL-2, IL-4, IL-9, IL-15 and IL-21. Whereas IL-7Rα expression is tightly regulated and almost exclusively found on lymphoid cells, γc is constitutively expressed by most hematopoietic cell types [1,2]. Mice with IL-7 or IL-7Rα deficiency show a block in T- and B-lymphocyte development, resulting in non-functional peripheral T-cells and reduced numbers of functional peripheral B-cells [3,4,5]. In humans, genetic alterations causing the loss-of-function of IL-7Rα or γc result in severe combined immunodeficiency through impaired thymocyte differentiation and T-cell survival, emphasizing the critical role of IL-7 signaling in T-cell development [6,7,8].

IL-7 is produced by stromal cells in lymphoid organs such as the bone marrow, thymus, spleen and lymph nodes, as well as in non-lymphoid tissues including the intestine, skin, lung and liver [9,10]. In contrast to other cytokines, the production of IL-7 occurs at a fixed rate, uninfluenced by external stimuli. The amount of available IL-7 is therefore dependent on the rate of consumption by lymphocytes rather than on the rate of production and, as such, plays a role in regulating lymphocyte homeostasis. In normal conditions, the amount of IL-7 is just sufficient to support the survival of a specific number of T-cells, with excess T-cells not being able to survive. After lymphocyte depletion, however, abundant IL-7 will stimulate lymphocyte proliferation until homeostasis is restored [1,11]. Contrary to the constitutive production of IL-7, the expression of *IL7R* is tightly regulated during lymphocyte development and, in mature T-cells, extremely influenced by external stimuli [1,12,13].

IL-7 signaling is initiated when binding of IL-7 to the IL-7R induces the heterodimerization of and conformational changes in IL-7Rα and γc (Figure 1). These conformational changes bring together the tyrosine kinases Janus kinase 1 (JAK1), associated with IL-7Rα, and JAK3, associated with γc, which phosphorylate each other, thereby increasing their kinase activity. Subsequently, the activated JAK proteins phosphorylate tyrosine residue Y449 in the cytoplasmic domain of IL-7Rα and, as such, create a docking site for Src homology-2 (SH2) domain-containing downstream effectors. One such critical effector is signal transducer and activator of transcription 5 (STAT5), which is phosphorylated on tyrosine residue Y694 by the JAK proteins upon docking to IL-7Rα. Phosphorylated STAT5 then homodimerizes and translocates to the nucleus, where it activates the expression of its target genes, such as *BCL2*, *CISH*, *MYC*, *OSM* and *PIM1*, which are involved in inhibiting apoptosis, as well as stimulating survival, cell cycle progression, proliferation and differentiation [1,14,15,16,17]. STAT5 is also phosphorylated on serine residues S725 and S779 by serine/threonine kinases, and this phosphorylation is required for full transcriptional activation of STAT5 [18]. Other effector molecules downstream of IL-7 signaling include STAT1 and STAT3, which play a role in, amongst others, T-cell homeostasis [19,20,21,22]. In addition to the JAK-STAT pathway, IL-7 signaling activates the PI3K-AKT and Ras-MAPK pathways [23,24]. While PI3K-AKT activation is initiated by the docking of PI3K to phospho-Y449, Ras-MAPK signaling is suggested to be activated via a cross-talk with the JAK-STAT pathway [10,25,26,27,28,29].

Physiologically, IL-7-initiated signaling is only transient, as negative regulation and termination of the signal is rapidly activated by (1) clathrin-dependent endocytosis and the subsequent proteasomal degradation of IL-7Rα, (2) dephosphorylation of JAK1, JAK3 and STAT5 by the protein tyrosine phosphatases (PTP) non-receptor type 2 (PTPN2) and receptor type C (PTPRC, also known as CD45), (3) the suppression of IL-7 signaling by the suppression of cytokine signaling (SOCS) proteins, and (4) the SUMOylation of STAT5 by protein inhibitors of STATs (PIAS) proteins [30,31].

In addition to its role in IL-7 signaling, IL-7Rα can heterodimerize with cytokine receptor-like factor 2 (CRLF2, also known as TSLPR), thereby forming the receptor for thymic stromal lymphopoietin (TSLP) [32,33]. The inclusion of IL-7Rα in both the IL-7 and TSLP receptor complex suggests that the ligand-driven dimerization of these two different receptors results in the activation of a common downstream signaling pathway. Indeed, signaling induced by both receptors activates STAT5 and upregulates the expression of STAT5 target genes [34]. However, the mechanisms underlying the transcriptional activation of STAT5 differ [35]. Whereas IL-7 signaling activates STAT5 via the phosphorylation of JAK1 and JAK3, TSLP-initiated signaling does so by the activation of JAK1 and JAK2 (Figure 2) [22,36].

In contrast to mouse Tslp signaling which promotes the proliferation and differentiation of pre-B-cells, peripheral CD4^+^ T-cells and myeloid dendritic cells (mDCs), in humans, TSLP only activates mDCs [37,38,39,40,41,42,43]. As such, via interaction between these activated mDCs and CD4^+^ T-lymphocytes, TSLP-initiated signaling is involved in the regulation of the positive selection of regulatory T-cells, maintenance of peripheral CD4^+^ T-cell homeostasis and induction of CD4^+^ T-cell-mediated allergic inflammation [43,44,45,46,47]. However, although in vitro and in vivo experiments suggest that TSLP may play a role in lymphocyte development, studies in mice and humans lacking the ability to respond to TSLP show that TLSP signaling is not essential for lymphopoiesis [6,48,49].

## 2. The Role of IL-7 Signaling in Acute Lymphoblastic Leukemia

The role of IL-7-induced signaling in the development of lymphoid malignancies has been suggested by several mouse models. IL-7 transgenic mice displayed accelerated mortality due to T- and B-cell lymphoma development and also AKR/J mice, which overexpress wild type IL-7Rα, spontaneously developed T-cell lymphoma [50,51,52,53]. Moreover, it was demonstrated in patient-derived xenograft (PDX) models that T-cell acute lymphoblastic leukemia (T-ALL) cells developed more slowly when engrafted in IL-7-deficient mice compared to mice expressing IL-7. In these models, IL-7 deficiency resulted in a decreased expansion of T-ALL cells in the bone marrow, reduced infiltration in the peripheral blood and extramedullary sites and delayed leukemia-associated death [54].

In addition, wild type IL-7Rα is detected on leukemic cells from more than 70% of patients with ALL, and these IL-7Rα cell surface expression levels correlated with IL-7 response in vitro [2,50,53,54,55,56]. Moreover, several studies have identified oncogenic mechanisms that increase IL-7Rα expression and cell surface levels [57]. For example, the transcription factor NOTCH1, which is activated by gain-of-function mutations in more than 65% of T-ALL, upregulates the expression of *IL7R* [58]. In early T-cell precursor ALL (ETP-ALL), the aberrant expression of the transcription factor *ZEB2* resulted in increased expression of *IL7R*, and ZEB2-induced *IL7R* upregulation promoted T-ALL cell survival in vitro and in vivo [59]. Furthermore, the arginine to serine substitution at residue 98 (R98S) of RPL10, a mutation identified in up to 8% of patients with T-ALL, was shown to increase the expression of *IL7R* and downstream signaling molecules [60]. Lastly, the reduced expression of SOCS5 was found in T-ALL patients with KMT2A translocations and resulted in the upregulation of IL-7Rα expression levels and the activation of JAK-STAT signaling, thereby promoting T-ALL cell proliferation in vitro and in vivo [61].

These results already show that the deregulated expression of both IL-7 and its receptor can contribute to the development of lymphoid malignancies, as well as that T-ALL cells often remain dependent on IL-7-induced signaling for survival, cell cycle progression and proliferation. Moreover, in the last few years, it has become increasingly clear that in lymphoid malignancies, many signaling molecules of the IL-7 pathway carry genetic alterations. Below, we discuss the role of the deregulation of the most important components of IL-7 signaling in lymphoid malignancies (Figure 3).

### 2.1. Mutations in the IL-7 Receptor

IL-7Rα is a type I transmembrane cytokine receptor consisting of an extracellular domain, a single transmembrane domain and an intracellular domain [62]. The extracellular domain contains two fibronectin type III-like domains (DN1 and DN2) with four paired cysteines and a juxtamembrane WSxWS motif which are involved in mediating the correct folding of the extracellular domain and, as such, binding of IL-7 to the IL-7R [63]. The intracellular region consists of a four-point-one protein, ezrin, radixin, moesin (FERM) domain, comprising a juxtamembrane BOX1 domain that is required for association with JAK1, as well as the tyrosine residue Y449 which, when phosphorylated, creates a docking site for STAT5 (Figure 3) [64].

Gain-of-function mutations in IL-7Rα are identified in up to 10% of T-ALL and about 2–3% of B-cell precursor ALL (B-ALL) cases [65,66,67,68,69,70]. In T-ALL, *IL7R* mutations are substantially enriched in immature T-ALL cases and in cases aberrantly expressing *TLX1, TLX3* or *HOXA*, and also in B-ALL *IL7R* alterations are found in specific subtypes, including Ph-like, CRLF2-rearranged, iAMP21-positive, IKZF1 mutant or PAX5 mutant B-ALL [57]. The *IL7R* mutations are heterozygous and almost always located in exon 6, where they introduce in-frame insertions or deletions–insertions in the extracellular juxtamembrane (EJ) or transmembrane (TM) domain of IL-7Rα. Strikingly, the majority of these alterations (>80%) introduce an unpaired cysteine (Figure 3) [64].

Several studies investigating the underlying mechanism of cysteine-introducing mutations have shown that the unpaired cysteine promotes the formation of *de novo* intermolecular disulfide bonds between mutant IL-7Rα chains resulting in constitutive IL-7Rα homodimerization and JAK1 and STAT5 phosphorylation, independent of IL-7, γc or JAK3 [24,65,66,67]. The expression of cysteine-introducing mutations in both the IL-7-dependent thymocyte cell line D1 and the IL-3-dependent pro-B cell line Ba/F3 was able to transform cells to cytokine-independent proliferation. Transformed cells were sensitive to the JAK inhibitors Pyridone 6, ruxolitinib and tofacitinib, as well as a STAT5-specific small molecule inhibitor [24,65,67,71]. Moreover, the intravenous injection of IL-7Rα p.L242-L243insNPC-expressing D1 cells, as well as IL-7Rα p.L242-L243insGC- and IL-7Rα p.IL241–242TC-positive Arf^-/-^ thymocytes into Rag^-/-^ mice resulted in leukemia development. Treatment with ruxolitinib in these models significantly reduced leukemic burden and prolonged survival [71,72]. These observations imply that cysteine-introducing mutations in IL-7Rα drive cellular transformation in vitro and leukemia development in vivo by activating downstream JAK-STAT signaling.

In addition to these insertions or deletion–insertions, there is also the recurrent point mutation IL-7Rα p.S185C that results in the introduction of an unpaired cysteine residue [66]. This mutation is exclusively identified in B-ALL and conferred IL-3-independent proliferation on Ba/F3 cells only upon co-expression with CRLF2 [66].

The non-cysteine *IL7R* mutations can be classified into two groups according to their localization in the EJ-TM or transmembrane region (TM) [64]. However, the exact mechanisms by which these non-cysteine mutations contribute to increased IL-7R signaling are not fully resolved yet. EJ-TM mutations, constituting 4% of *IL7R* mutations, mainly introduce a positively charged amino acid (i.e., arginine, histidine or lysine) in IL-7Rα, which can engage in electrostatic interactions with a negatively charged residue present in wild type γc [24,73]. These alterations are thus suggested to facilitate IL-7Rα-γc heterodimerization and increase IL-7 sensitivity. On the other hand, TM mutations (9%) insert residues which generate de novo homodimerization motifs (such as ExxxV, SxxxA and SxxxG) which are suggested to form intermolecular hydrogen bonds, thereby stabilizing IL-7Rα homodimers and inducing IL-7-independent signaling [74,75,76].

These results illustrate that there are different mechanisms by which activating IL-7Rα mutations contribute to the increased activation of the IL-7 signaling pathway and that a variety of oncogenic effects can be expected, ranging from a slight increase in IL-7 sensitivity to constitutive IL-7-independent signaling. Moreover, as IL-7 is only available at limited amounts in the thymus and in other lymphoid tissues, leukemia cells that become increasingly sensitive to IL-7 have a survival and proliferative advantage compared with wild type lymphoid cells as a result of increased IL-7 signaling pathway activation, most likely explaining why in many T-ALL cases that carry a mutation in *IL7R*, additional alterations in *JAK1*, *JAK3*, *PTPN2*, *PTPRC* and/or *DNM2* are found, as described in the following sections. By accumulating multiple mutations in the IL-7 signaling pathway, cells eventually become (almost) completely independent of IL-7.

In contrast to IL-7Rα, which is recurrently mutated in ALL as described above, activating alterations in the other IL-7 receptor chain, γc (Figure 1) are extremely rare. In fact, no such alterations have been described in ALL, but some were identified in T-cell prolymphocytic leukemia (T-PLL), a rare T-cell leukemia [77]. Although rare, these mutations further illustrate the importance of deregulated IL-7 signaling in various lymphoid malignancies and are in line with the major role for IL-7 in regulating T-cell survival and proliferation.

### 2.2. Mutations in the JAK1 and JAK3 Tyrosine Kinases

The Janus kinase (JAK) family comprises four members of non-receptor tyrosine kinases (JAK1, JAK2, JAK3 and TYK2) which associate with cytokine receptors that lack intrinsic kinase activity to mediate cytokine-induced signaling. All JAK family members share a common structure consisting of an N-terminal FERM domain and SH2-like domain which are required for associating the JAK proteins to cytokine receptors, a C-terminal pseudokinase (JAK homology 2, JH2) and kinase domain (JH1) [78,79]. The catalytically inactive JH2 pseudokinase domain functions as a negative regulator of the JH1 kinase domain, as it stabilizes JH1 in an inactive conformation in the absence of cytokines (Figure 3) [80,81,82]. In contrast to their conserved structure, the different JAK family members preferentially associate with specific cytokine receptors that are expressed by specific cell types and, as such, facilitate difference in function [83]. At the IL-7R, JAK1 associates with IL-7Rα, whereas JAK3 binds to γc (Figure 1).

Mutations in *JAK1* and *JAK3* are most frequent in T-ALL and T-PLL and are mainly found in the JH2 pseudokinase and JH1 kinase domains (Figure 3) [67,83,84,85,86,87,88,89,90,91,92]. The prevalence of *JAK1* and *JAK3* mutations differs substantially between studies, dependent on the number and age of the patients included [67,83,84,85,86,87,88,90,91,92,93]. JAK3 is the most frequently mutated component of the IL-7 signaling pathway, with mutations identified in up to 16% of T-ALL, while JAK1 alterations are rather rare and mainly found in cases that also carry *JAK3* or *IL7R* mutations. Moreover, if activating *JAK1* mutations do occur, they are usually found in subclones, rather than in the major clone. It is surprising that JAK1 is less frequently mutated than JAK3, since previous studies demonstrated that JAK1 is the most important kinase of the IL-7 signaling pathway [94]. This could suggest that extremely strong activation of the IL-7 signaling pathway is not preferred, as it could result in cell exhaustion and/or other undesirable side effects.

The transforming capacity of JAK3 mutations has been investigated using in vitro cell-based assays and in vivo bone marrow transplant models [84,94,95,96]. Although the majority of JAK3 mutations in ALL are located in the JH2 pseudokinase and JH1 kinase domain, the most frequent alteration, JAK3 p.M511I, affects an amino acid right outside the pseudokinase domain [67,89,90,97]. Moreover, not all mutations identified are gain-of-function alterations that drive leukemogenesis. So-called passenger mutations, such as some JAK3 kinase domain mutations and the majority of mutations in the FERM/SH2 domain of JAK3, do not contribute to cellular transformation nor leukemia development [84,94,95,96].

Reconstitution of the IL-7R in HEK293T cells showed that the JAK3 pseudokinase domain mutants required a functional receptor complex consisting of IL-7Rα and γc to constitutively phosphorylate and activate STAT5 [94,95]. In contrast, this was not the case for the JAK3 kinase domain mutants p.L857P and p.L857Q. In line with these observations, it was demonstrated that JAK3 pseudokinase mutations signal through JAK1 and that JAK1 kinase activity is required for their oncogenic properties [94]. These differences in functional IL-7R requirement and downstream JAK kinase activity are important to determine which JAK inhibitors can be used to target leukemia cells carrying a certain JAK3 mutation. Ruxolitinib, which is a JAK1/JAK2-selective inhibitor, inhibited the proliferation of cells transformed by pseudokinase domain JAK3 mutants, whereas cells expressing JAK3 kinase domain mutations were less sensitive [94,95]. The latter were, in contrast, more sensitive to JAK3-specific inhibitors [84,95,96]. For the JAK1/JAK3-specific inhibitor tofacitinib, the pseudokinase and kinase domain mutants showed a similar sensitivity, whereas treatment with a combination of ruxolitinib and tofacitinib synergistically inhibited the proliferation of cells transformed by pseudokinase, but not kinase domain mutants [94,95].

Mouse bone marrow transplant models of different JAK3 mutants resulted in the development of lymphoid malignancies with an average disease latency of about 150 days [94]. Interestingly, these lymphoid malignancies showed slight differences in phenotype. That is, while pseudokinase domain mutants homogenously induced a T-ALL-like disease, the expression of kinase domain mutations resulted in more heterogenous phenotypes with various lympho- and myeloproliferative malignancies [94]. In addition, a recent study by de Bock and colleagues showed that co-expression of the JAK3 mutant p.M511I and HOXA9 substantially reduced disease latency to about 40 days as a result of strong oncogenic cooperation [98].

In up to a third of T-ALL patients carrying a *JAK3* mutation, mutant JAK3 signaling is further enhanced by either the loss of wild type JAK3 or the acquisition of a secondary JAK3 mutation [99]. Degryse et al. showed that wild type JAK3 competed with JAK3 pseudokinase domain mutants for binding to the IL-7R and, as such, suppressed its transforming capacity. Moreover, acquiring a second mutation in the mutant JAK3 allele increased downstream JAK-STAT signaling, as shown by increased STAT5 phosphorylation.

The most extensively studied genetic alterations in *JAK1* are the pseudokinase domain mutations p.A634D, p.V658F and p.S646F [56,81,82,83,85,87,100]. These three mutants, as well as all other pseudokinase and kinase domain mutants studied, were able to transform Ba/F3 cells to IL-3-independent proliferation by constitutively phosphorylating STAT5. In the BM5247 T-lymphoma cell line, the expression of JAK1^A634D^ resulted in strong JAK1 and STAT5 phosphorylation and, as such, protection from dexamethasone-induced apoptosis [83]. Moreover, the three JAK1 pseudokinase mutants p.A634D, p.V658F and p.S646F were able to constitutively phosphorylate and activate STAT proteins when expressed in HEK293T and/or U4C cells in the absence of any other receptor complex component [81,82,83,87]. Hornakova et al., however, showed that the recruitment and docking of both JAK1 and STAT5 to a functional alpha chain were required for the alpha chain-mediated constitutive activation of STAT proteins by JAK1 pseudokinase domain mutants [82,100].

The proliferation of JAK1^S646F^-transformed Ba/F3 cells was inhibited by ruxolitinib and also a PDX model of JAK1 mutant B-ALL showed sensitivity to ruxolitinib [56,85,87]. Overall, these results together with the results on IL-7Rα and JAK3 indicate that ruxolitinib, which is currently used to treat JAK2 mutant malignancies, may also be a promising drug for the treatment of *IL7R*, *JAK1* or *JAK3* mutant cases.

### 2.3. Mutation in the STAT Family Member STAT5B

Signal transducer and activator of transcription 5B (STAT5B) belongs to the STAT family of transcription factors, which play a critical role in cytokine receptor signaling. All STAT family members have a common structure. The N-terminal domain is required for interaction with co-activators as well as higher-order interactions between activated STAT5 dimers. The central DNA-binding domain, which is involved in the recognition of the specific DNA binding sequence, is coupled to the SH2 domain by a flexible linker. This SH2 domain recognizes phosphotyrosine residues and plays a critical role in the recruitment of STATs to activated cytokine receptors, the interaction of STAT family members with JAK proteins and the dimerization of phosphorylated STAT proteins. Between the SH2 domain and the transactivation domain resides a conserved tyrosine residue (Y694) whose phosphorylation is essential for the activation and dimerization of all STAT family members. The C-terminal transactivation domain is required for coordinating the transcriptional machinery and contains two serine residues whose phosphorylation is required for full transcriptional activity (Figure 3) [101].

Gain-of-function alterations in *STAT5B* are identified in 6% of pediatric and up to 9% of adult T-ALL and are located in both the SH2 and transactivation domain. The most prevalent *STAT5* alteration is the p.N642H SH2 domain mutation (Figure 3) [102,103]. In T-PLL, a similar distribution of STAT5B mutations is observed [77].

In vitro and ex vivo cell-based assays showed that SH2 domain mutations and, to a lesser extent, transactivation domain mutations resulted in increased STAT5B Y694 phosphorylation and the upregulation of STAT5 target genes [77,102,103]. Moreover, the expression of STAT5B^N642H^ was able to transform cells to cytokine-independent proliferation [77,103].

Interestingly, Ba/F3 cells transformed by the expression of STAT5B^N642H^ showed sensitivity to ruxolitinib and tofacitinib, and treatment with these JAK inhibitors resulted in decreased STAT5B phosphorylation and reduced STAT5B target gene expression [104]. These in vitro observations imply that STAT5B^N642H^ induces JAK kinase activity, and that this activation is required for the phosphorylation and transcriptional activation of STAT5B. Indeed, also in vivo experiments using a STAT5B^N642H^ transgenic mouse model illustrated that the transforming capacity of STAT5B^N642H^ depends on phosphorylation by JAK1, as the leukemic burden was substantially reduced upon treatment with ruxolitinib [105]. In contrast, Kontro and colleagues observed that leukemic cells of a T-ALL patient carrying three STAT5B mutations did not show sensitivity to ruxolitinib or tofacitinib when treated ex vivo, suggesting that the co-occurrence of these three mutations constitutively activated STAT5B, independent of upstream JAK kinase activity [102].

### 2.4. Inactivation of the Protein Tyrosine Phosphatases PTPN2 and CD45

Protein phosphatases can be classified into two major families based on their substrate specificity: the protein tyrosine phosphatases (PTPs) and the serine/threonine phosphatases. The 107 human PTP genes are subdivided into four classes (Class I–IV) based on the amino acid sequence in the catalytic domain. Class I is the largest and can be classified into classical PTPs and dual specificity PTPs, and the former can be further subdivided into receptor PTPs and non-receptor PTPs. Both Class I and Class II PTPs have been shown to negatively regulate and terminate JAK-STAT signaling by dephosphorylating both JAK and STAT molecules [106].

PTP non-receptor type 2 or PTPN2 (also known as TC-PTP) is a ubiquitously expressed non-receptor PTP that exists as two splice variants [107]. Both variants consist of an N-terminal catalytic PTP domain followed by a C-terminal domain which includes either a nuclear localization signal or an ER targeting sequence (Figure 3) [106]. Although PTPN2 deficiency increases JAK-STAT signaling in various cell types, loss-of-function genetic alterations in *PTPN2* have been identified mainly in TLX1-expressing T-ALL cases [107,108]. These loss-of-function alterations typically involve mono- or bi-allelic deletions of the entire *PTPN2* gene [107].

The mechanism by which *PTPN2* deletion contributes to T-ALL is assigned to the negative regulation of JAK-STAT signaling [107,108,109]. In T-ALL, the expression of TLX1 is typically associated with gain-of-function mutations in the IL-7 signaling pathway, as well as with ABL1 fusion proteins, which both result in the constitutive activation of STAT5. Kleppe et al. found a direct cooperation between the loss of PTPN2 and oncogenic kinases involved in IL-7R-JAK-STAT signaling, such as mutant JAK1 and the NUP214-ABL1 fusion protein [107,109].

Another negative regulator of IL-7 signaling is the PTP receptor type C (PTPRC, also known as CD45) which is a classical receptor PTP that exists as several splice variants that are variably expressed by the majority of hematopoietic cells [110]. CD45 is a critical positive regulator of T- and B-cell receptor-mediated signaling [111,112,113], as well as a negative regulator of members of the JAK family via direct dephosphorylation or by adaptor protein recruitment [114,115]. The loss of CD45 expression has been observed in up to 4% of pediatric T-ALL and around 13% of pediatric B-ALL [116]. Moreover, Porcu et al. identified *CD45* inactivating alterations in patients with T-ALL, resulting in low CD45 expression or the loss of CD45 phosphatase activity [117]. Interestingly, these CD45 mutations all co-occurred with activating mutations in the IL-7R-JAK-STAT pathway and *CD45* knockdown experiments showed the increased activation of JAK-STAT signaling downstream of mutant IL-7Rα or JAK1 [117].

### 2.5. Alterations in DNM2

Dynamin 2 (DNM2), a ubiquitously expressed large GTPase, plays an essential role in clathrin-dependent endocytosis (CDE), a process that regulates receptor signaling as well as the recycling and degradation of receptor molecules [118]. During CDE, ligand-bound receptors, such as IL-7-bound IL-7R, are recruited to clathrin-coated pits in the cell membrane which then invaginate to form budding vesicles. Subsequently, GTP-dependent constriction by DNM2 results in the formation of clathrin-coated endocytic vesicles [118]. DNM2 contains five distinct domains, including the N-terminal GTPase domain and the C-terminal GTPase effector domain, that each impart a specific function during CDE (Figure 3) [119].

Genetic alterations in *DNM2* are identified in around 10% of adult T-ALL and up to 20% of the ETP subtype of T-ALL and are heterozygous, with frameshifts, non-sense, missense and splice mutations and deletions throughout the whole gene [67,120].

The mechanism by which alterations in *DNM2* promote leukemogenesis was elucidated by Tremblay and colleagues using a Lmo2^Tg^Dnm2^V265G^ transgenic mouse model [118]. They observed that this mutation in *Dnm2* cooperated with Lmo2 expression to accelerate the development of T-ALL. DNM2 loss-of-function impaired the formation of clathrin-coated endocytic vesicles and blocked the internalization of IL-7R, which led to enhanced IL-7 signaling in preleukemic thymocytes. In agreement with this, *DNM2* mutations co-occur with additional activating alterations in the IL-7 signaling pathway, suggesting a cooperation between these genetic alterations in leukemia development [67,108].

### 2.6. Alterations in the CRLF2 Receptor Chain

As described above, IL-7Rα can also form heterodimers with CRLF2 (Figure 2), a type I transmembrane cytokine receptor with a unique conformation and only one single tyrosine residue at the C-terminus (Figure 3) [35]. This heterodimer forms the receptor for TSLP.

Gain-of-function alterations in *CRLF2* are identified in about 5% of pediatric and adult B-ALL overall and in up to 60% of B-ALL arising in patients with Down syndrome, but so far not in any other lymphoid malignancy [121,122,123,124,125]. The most common genetic abnormalities involving CRLF2 result in its upregulation due to chromosomal rearrangements at the pseudoautosomal region 1 (PAR1) of chromosome X or Y or translocations of the CRLF2-containing PAR1 with the IGH@ locus [121,123,124,125].

The overexpression of CRLF2 was able to enhance the proliferation of early B-cell precursor cells in vitro [121]. However, *CRLF2* knockdown in the B-ALL cell line MUTZ5 only partially abrogated cell proliferation and CRLF2 expression in primary bone marrow progenitor cells did not result in leukemia development in vivo [121,126]. Together, these observations imply that the aberrant expression of *CRLF2* is not sufficient to drive malignant transformation. Indeed, in the majority of CRLF2-overexpressing ALL patients, additional mutations in the TSLP signaling pathway are identified. For example, around half of ALL patients overexpressing CLRF2 carry gain-of-function mutations in *JAK2* [126,127], indicating that these two genetic alterations may cooperate to promote leukemogenesis [125].

An activating mutation in *CRLF2*, which results in the substitution of the phenylalanine residue F232 to an unpaired cysteine, has also been identified in CRLF2-overexpressing B-ALL (Figure 3) [123,124]. Similar to the cysteine-introducing alterations in *IL7R*, the unpaired cysteine residue is introduced in the EJ-TM region and promotes the formation of de novo intermolecular disulfide bonds between mutant chains [123]. As such, this p.F232C mutation results in the spontaneous homodimerization of CRLF2^F232C^ and constitutive phosphorylation and activation of JAK2 and STAT5, independent of TSLP or IL-7Rα [123]. Moreover, the expression of CRLF2^F232C^ was able to transform cytokine-dependent cell lines to cytokine-independent proliferation in vitro, suggesting a role of *CRLF2* gain-of-function alterations in malignant transformation [123,124,126,128].

## 3. The Role of IL-7 Signaling in Other Lymphoid Malignancies

The deregulation of the IL-7 signaling pathway is frequently observed in T-ALL and T-PLL, as well as in other T-cell malignancies in particular and lymphoid malignancies in general. Indeed, stimulation with IL-7 promoted the proliferation of cutaneous T-cell lymphoma (CTCL) cells, and also Sézary lymphoma cells were sensitive to IL-7 [129,130]. In addition, IL-7 signaling is suggested to be involved in chronic lymphoid leukemia (CLL) and Hodgkin’s lymphoma, and stimulation of CLL cells with IL-7 resulted in increased proliferation in vitro [131,132].

Furthermore, gain-of-function alterations in the IL-7 signaling pathway were identified in the majority of T-cell lymphomas (reviewed by Waldmann and colleagues) [133]. Similar to ALL, mutations in JAK1 and JAK3 are mainly found in the pseudokinase domain and in STAT5B the p.N642H SH2 domain mutation frequently occurs [133]. Activating mutations in *JAK1* are found in CTCL, natural killer cell lymphoma (NKCL) and large granulocytic leukemia (LGL), as well as in 20% of ALK-negative anaplastic large cell lymphoma (ALCL), and treatment with ruxolitinib reduced tumor growth in an ALK-negative ALCL PDX model in vivo [133,134]. About one third of patients with NKCL carry *JAK3* gain-of-function alterations, and a NKCL PDX model was sensitive to tofacitinib, with delayed tumor growth upon treatment [135,136]. *JAK3* mutations are also found in CTCL, LGL, peripheral T cell lymphoma not otherwise specified (PTCL-NOS) and human T cell lymphotropic virus 1-associated adult T cell leukemia/lymphoma [133]. The STAT5B p.N642H gain-of-function mutation, as well as other SH2 domain mutations, were identified in CTCL, NKCL, LGL, enteropathy-associated T cell lymphoma and γδ T cell lymphoma and, like in ALL, resulted in increased STAT5B phosphorylation and transcriptional activity [133,137,138]. Moreover, in addition to TLX1^+^ T-ALL, bi-allelic deletions of the entire *PTPN2* gene locus were identified in the Hodgkin’s lymphoma cell line SUP-HD1 and in 2 out of 39 patients with PTCL-NOS [139].

## 4. Therapeutic Targeting of the IL-7 Signaling Pathway in Lymphoid Malignancies

The gain-of-function alterations in the IL-7 signaling pathway provide new therapeutic targets for the treatment of ALL and other lymphoid malignancies [2,140]. In B-ALL, IL-7Rα expression is directly correlated with central nervous system (CNS) involvement at diagnosis, and the treatment of PDX models with a commercially available mouse antibody targeting IL-7Rα was able to substantially reduce leukemic cell infiltration in the CNS and prolong survival [141]. In addition, the delivery of a cytotoxic agent using a mouse anti-IL-7Rα antibody efficiently eliminated IL-7-induced glucocorticoid-resistant cells in a syngeneic mouse model [142]. Recently, Akkapeddi et al., and Hixon and colleagues developed human and chimeric mouse–human monoclonal antibodies targeting IL-7Rα, which induced antibody-dependent cell-mediated cytotoxicity against T-ALL cells in vitro and were effective for treating T-ALL in PDX models of both established and relapsed disease in vivo [143,144]. A phase I clinical trial already suggested that these therapeutic antibodies are well tolerated in healthy volunteers, and their efficacy will be further investigated in patients with T-ALL [57]. Treatment with the reducing agent N-acetylcysteine was able to inhibit spontaneous disulfide bond formation between mutant IL-7Rα chains and, as such, reduced constitutive IL-7R signaling, thereby promoting apoptosis of IL-7Rα mutant cells in vitro and reducing leukemic burden in vivo [145].

Another therapeutic strategy is to target downstream signaling molecules and/or target genes. The selective JAK1/JAK2 inhibitor ruxolitinib, which is FDA approved for the treatment of myelofibrosis and polycythemia vera, as well as other small molecule JAK inhibitors showed efficacy in pre-clinical studies using in vitro and in vivo models of T-cell malignancy and B-ALL [24,56,65,67,70,71,72,85,94,95,104,105,146]. Phase I/II and phase II/III clinical trials with ruxolitinib for the treatment of ALL are currently ongoing, and given the therapeutic benefit, St. Jude Children’s Research Hospital has recently incorporated ruxolitinib into the induction therapy of ETP-ALL (NCT03117751) [57]. Moreover, gain-of-function alterations in the IL-7 pathway also result in the activation of PI3K-AKT and Ras-MAPK signaling, and although treatment with small molecule inhibitors targeting PI3K, AKT or MEK alone were not effective, inhibiting both the PI3K-AKT and Ras-MAPK pathway synergistically reduced the cytokine-independent proliferation of Ba/F3 cells expressing mutant IL-7Rα, JAK1 or JAK3, as well as primary T-ALL cells [24].

IL-7 signaling results in the upregulation of STAT5 target genes, including *BCL2* and *PIM1*, which are required for IL-7-mediated T-ALL cell survival [17]. Primary patient-derived T-ALL cells carrying activating *JAK3* mutations showed increased sensitivity, ex vivo and in vivo, to combination treatment with the selective JAK1/JAK3 inhibitor tofacitinib and the selective BCL2 inhibitor venetoclax than to one of the inhibitors alone [147]. Similar results were obtained for in vitro and in vivo treatment of the D1 thymocyte cell line expressing mutant IL-7Rα with ruxolitinib and venetoclax [71]. Treatment with both ruxolitinib and a selective PIM1 inhibitor synergistically reduced the proliferation of an IL-7Rα mutant T-ALL cell line in vitro and leukemic burden in a PDX model of JAK3 mutant T-ALL in vivo [98].

## 5. Conclusions

The IL-7 signaling pathway is critical for normal lymphoid development and it is therefore not surprising that this pathway is deregulated in various lymphoid malignancies. Perhaps the most important biological lesson to learn from all these studies is that, typically, more than one gain-of-function genetic alteration is present in the IL-7 signaling pathway, indicating that a strong control mechanism is present, which cancer cells are able to overcome by acquiring multiple mutations. Considering clinical applications, these studies have taught us that JAK1 plays a central role in the mutant IL-7 signaling pathway, and that JAK1 inhibitors such as ruxolitinib could play a role in further improving the treatment of lymphoid malignancies with IL-7 signaling pathway mutations.

## Figures and Tables

**Figure 1 pharmaceuticals-14-00443-f001:**
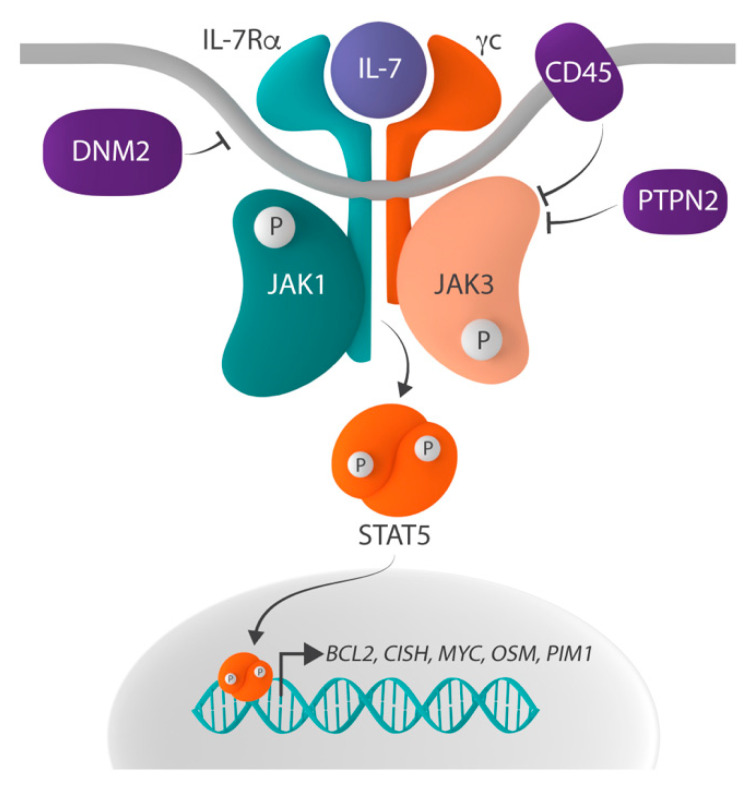
Schematic representation of the IL-7R-JAK-STAT signaling pathway. The IL-7R-JAK-STAT signaling pathway is activated when interleukin-7 (IL-7) binds to the IL-7 receptor (IL-7R), which consists of IL-7Ralpha (IL-7Rα) and the common gamma chain (γc), resulting in the phosphorylation and thus activation of Janus kinase 1 (JAK1) and JAK3. Activated JAK proteins phosphorylate signal transducer and activator of transcription 5 (STAT5), and phosphorylated STAT5 homodimerizes and translocates to the nucleus where it regulates the expression of STAT5 target genes, such as *BCL2*, *CISH*, *MYC*, *OSM* and *PIM1*. Negative regulators of the pathway include the protein tyrosine phosphatases (PTP) non-receptor type 2 (PTPN2) and receptor type C (PTPRC, also known as CD45), as well as the large GTPase dynamin 2 (DNM2) which plays a role in the clathrin-dependent endocytosis of IL-7R.

**Figure 2 pharmaceuticals-14-00443-f002:**
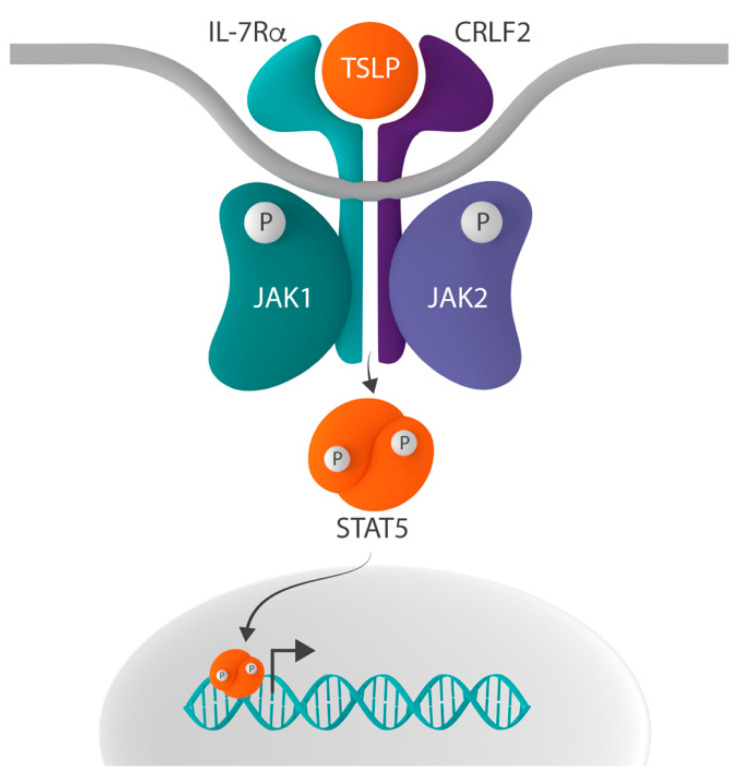
Schematic representation of the CRLF2-JAK-STAT signaling pathway. In addition to the common gamma chain, interleukin-7 receptor alpha (IL-7Rα) can form heterodimers with cytokine receptor-like factor 2 (CRLF2), thereby creating the receptor for thymic stromal lymphopoietin (TSLP). Whereas IL-7-induced signaling activates signal transducer and activator of transcription 5 (STAT5) via the phosphorylation of Janus kinase 1 (JAK1) and JAK3, the binding of TSLP to the TSLP receptor results in STAT5 activation via the phosphorylation of JAK1 and JAK2.

**Figure 3 pharmaceuticals-14-00443-f003:**
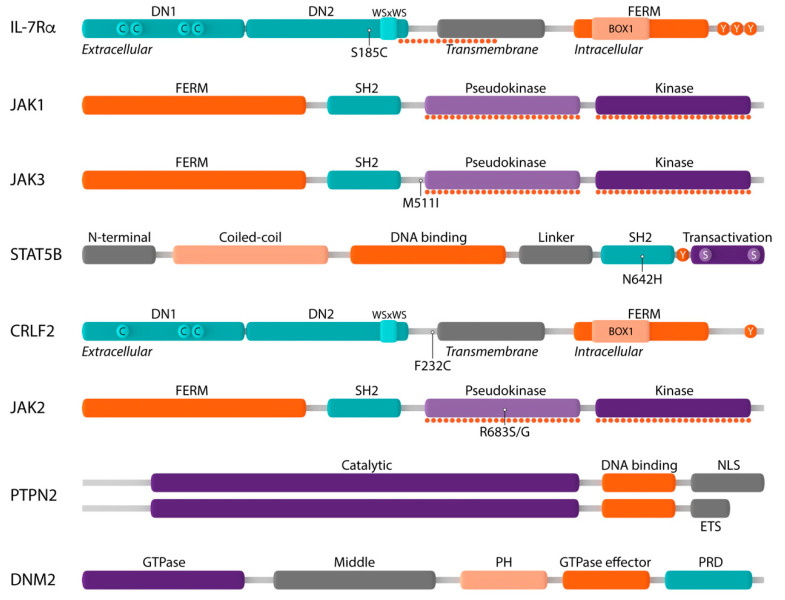
Schematic representation of components of the IL-7R-JAK-STAT and CRLF2-JAK-STAT signaling pathways and their main protein domains. Interleukin-7 receptor alpha (IL-7Rα): extracellular domain with fibronectin type III-like domains DN1 and DN2 and four paired cysteine residues (C) and WSxWS motif, transmembrane domain, intracellular domain with four-point-one protein, ezrin, radixin, moesin (FERM) domain, BOX1 domain and three tyrosine residues (Y); Janus kinase 1 (JAK1): FERM domain, Src homology-2 (SH2) domain, pseudokinase domain, kinase domain; JAK3: FERM domain, SH2 domain, pseudokinase domain, kinase domain; signal transducer and activator of transcription 5B (STAT5B): N-terminal domain, coiled–coil domain, DNA binding domain, linker domain, SH2 domain, tyrosine residue Y694 (Y), transactivation domain; cytokine receptor-like factor 2 (CRLF2): extracellular domain with fibronectin type III-like domains DN1 and DN2 and only three cysteine residues (C) and WSxWS motif, transmembrane domain, intracellular domain with FERM domain, BOX1 domain and only one tyrosine residue; JAK2: FERM domain, SH2 domain, pseudokinase domain, kinase domain; protein tyrosine phosphatase non-receptor type 2 (PTPN2): catalytic domain, DNA binding domain, nuclear localization signal (NLS) or ER targeting sequence (ETS); dynamin 2 (DNM2): GTPase domain, middle domain, plekstrin homology domain, GTPase effector domain, proline-rich domain. Mutational hotspots are shown by dotted lines and the most frequently occurring genetic alterations are indicated.

## Data Availability

Not applicable.

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
