# Peer review of "Deregulation of the Interleukin-7 Signaling Pathway in Lymphoid Malignancies"

_pharmaceuticals, 2021, doi:10.3390/ph14050443_

Round 1
Reviewer 1 Report
The manuscript entitled “Deregulation of the interleukin-7 signaling pathway in lymphoid malignancies” by Inge Lodewijckx and Jan Cools presents an extensive review of the literature about the IL7R pathway, which is often deregulated in lymphoid malignancies. It presents the biological function of IL7R, the signaling pathways involved in normal and malignant cells, and how this pathway can be a source of therapeutic targets for patients. It is very well written and extremely informative. It was a pleasure to read.
I only have two minor comments:
1. There is a typo: “Trembley” on line 381. It should be changed to “Tremblay”.
2. It would be interesting to include a small paragraph in the first section to mention which cells are responsible for the production of IL7.
Reviewer 2 Report
The manuscript titled Deregulation of the interleukin-6 signaling pathway in lymphoid malignancies is a well-written and thorough overview of the role of various IL-7 signaling molecules in normal and malignant lymphopoiesis.
This review will be a valuable reference for those in the fields of both IL-7 signaling and lymphoid malignancies and is suited for publication in Pharmaceuticals. It is my opinion that this manuscript can be accepted essentially as is, subject to the very minor comments listed below.
1. Over lines 162 - 164 you state “The IL7R mutations are heterozygous and almost always located in exon 6, where they introduce in-frame insertions of deletions-insertions in or right outside the transmembrane domain”. For clarification, it would be useful to indicate whether these insertions occur in the intracellular or extracellular portions of the receptor, or both. Although you state over the following sentence the majority of these insertions occur in the extracellular juxta membrane it is unclear whether this is the case for the other 20%.
2. Over lines 218-221 you state “All JAK family members share a common structure consisting of an N-terminal FERM domain which is involved 2 in associating the JAK proteins to cytokine receptors, an SH2-like domain and a C-terminal pseudokinase (JAK homology 2, JH2) and kinase domain (JH1).” The FERM and SH2 domains are together responsible for binding to receptor, as such this sentence should be amended to reflect this.
3. The sentence over lines 241-243 requires a reference. Sentence: “Although the majority of JAK3 mutations in ALL are located in the JH2 pseudokinase and JH1 kinase domain, the most frequent alteration, JAK3 p.M511I, affects an amino acid right outside the pseudokinase domain.”
